# Temperature Field Analytical Solution for OGFC Asphalt Pavement Structure

**Lin Qi [1], Baoyang Yu [2,3,\*], Zhonghua Zhao [1] and Chunshuai Zhang [2]**

[1] School of Civil Engineering, Shenyang Urban Construction University, Shenyang 110167, China; qilin6126@126.com (L.Q.); zzhua218@163.com (Z.Z.)

[2] School of Transportation and Geomatics Engineering, Shenyang Jianzhu University, Shenyang 110168, China; zcs2928448184@stu.sjzu.edu.cn

[3] Transportation Engineering College, Dalian Maritime University, Dalian 116026, China

\* Correspondence: yubaoyang12380@126.com; Tel.: +86-024-24694351

**Abstract:** The change law of the temperature field of an open-graded friction course (OGFC) asphalt pavement was studied. The thermal conductivity of OGFC asphalt mixtures with different oil–stone ratios was measured using a thermal-conductivity tester. The relationship between the oil–stone ratio and thermal conductivity was established, which was then used as the boundary condition of the temperature field. Using mathematical and physical methods based on thermodynamics and heat-transfer principles, an analytical solution of the temperature field of the OGFC asphalt pavement structure was developed. Data from an outdoor test of large Marshall specimens were compared with the analytical solution of the temperature field to verify the correctness of the model. The results show that the analytical model of the OGFC asphalt pavement structure temperature field can predict the temperature changes at different oil–stone ratios, times, and depths (from the road surface). The differences between the predicted results and test data at 0.01, 0.02, and 0.03 m from the road surface were 0.5, 0.7, and 0.9 °C, respectively, confirming that this study can be used to provide reference information for the design of OGFC asphalt pavement structures.

**Keywords:** pavement structure; open-graded friction course; thermal conductivity; temperature field; analytical solution





## 1. Introduction

The study of the temperature field of a pavement structure is an important part of pavement structural design, with research on the temperature field being indispensable for understanding temperature-related pavement defects [1]. An open-graded friction course (OGFC) asphalt pavement has a large void structure, and rainwater can be discharged quickly from the pavement through the voids during rainfall. Moreover, OGFC asphalt pavements have the advantages of gap connectivity, strong water permeability, and low noise. The pavement surface is rough, has anti-skid characteristics, and has good diffuse reflectance to light, which improves driving safety and is applicable to the wearing layer and upper layer of the pavement [2–4]. Additionally, OGFC asphalt pavements have more voids than other pavements. This improves airflow, which can make them susceptible to changes in air, temperature, and other external environmental factors [5]. In addition, sunlight, air, and water can easily enter the OGFC interior, accelerating the asphalt aging process and causing durability problems [6]. The large voids in OGFC asphalt pavements can lead to a high volume of air within the pavement structure, which can cause the thermal conductivity of the OGFC asphalt mixture to fluctuate considerably, having a major effect on the temperature field. Consequently, research on the temperature field of OGFC asphalt pavement structures is of great importance for the prevention of temperature-related defects and understanding the defect-generation mechanism in them [7,8]. In this study, the OGFC

asphalt pavement was selected as the research object to investigate the influencing factors and change rules of the temperature field of the pavement structure.

There have been several studies on the temperature field of asphalt pavements; their methods can be divided into three broad categories: mathematical statistics, numerical analysis, and theoretical analysis [9]. The theoretical-analysis method is based on meteorological data and the thermophysical parameters of pavement materials. This method applies heat-transfer principles and relevant assumptions and considers the natural environmental boundary conditions to solve the analytical expression of pavement temperature. Although this method requires a large volume of meteorological data and many pavement thermophysical parameters, it has strong adaptability and is not subject to geographical restrictions [10]. As early as 1957, Barber [11] regarded a pavement structure as a semi-infinite body and assumed that climate variables and temperature change sinusoidally, thereby establishing a relationship between the daily climate data and maximum pavement temperature. This study marked the beginning of the use of the theoretical-analysis method to analyze pavement temperatures. In 1970, Dempsey and Thompson [12] established a heat-transfer model for the evaluation of frost and temperature-related effects in a multi-layer pavement system. The model was derived from one-dimensional, forward finite-difference, heat-transfer theory and programmed for a computer solution. It was used to evaluate the temperature ranges of different pavement systems at different geographical locations. In 1972, Christison and Anderson [13] obtained the minimum surface temperature, minimum air temperature, surface-temperature amplitude, and air-temperature amplitude of pavements using actual measurements of two test sections in southwest Canada. They obtained linear relationships using regression analysis. Additionally, they used the basic theory of heat transfer to establish a pavement temperature-field prediction model based on a one-dimensional heat-transfer equation and the finite-difference method. In 1993, Lytton et al. [14] proposed a comprehensive impact model to reflect the impact of environmental factors on pavement structures. The model was used to simulate the influence of different environmental factors on the pavement temperature field and predict the temperature distribution of the pavement. The maximum and minimum pavement temperatures were estimated by inputting the pavement parameters and meteorological data for each layer of materials within the pavement structure. In the same year, Solaimanian and Kennedy [15] proposed a simple analytical equation to predict the maximum road-surface temperature based on the maximum temperature and hourly solar radiation. In 1998, Liang and Niu [16] calculated an analytical solution for the temperature field of a three-layer pavement structure using simplified boundary conditions, that is, without considering the influence of solar radiation and only applying the convective heat-transfer boundary conditions between the air and road surface. In 2000, Hermansson [17] designed a simulation model to calculate the temperature of asphalt concrete in summer. The heat-transfer equation was developed using the finite-difference method, and the temperature beneath the surface was calculated. The effectiveness of the radiation formula and the entire simulation model was verified via a comparison with measured results.

In 2002, Mrawira and Luca [18] proposed an improved test procedure for measuring the thermal conductivity of asphalt concrete pavements. Based on the measured thermal conductivity and thermal diffusivity, the incremental recursive method was used to analyze the transient temperature response of the road surface when the environment changed. In 2006, Diefenderfer et al. [19] established a daily maximum- and minimum-temperature prediction model by monitoring the actual pavement-surface temperature. The project's aim was to study the impact of temperature on pavement performance. In 2012, Wang [20] and Wang and Roesler [21] developed a one-dimensional temperature-prediction algorithm for multi-layer pavement systems based on the measured temperature data of road surfaces using Laplace and inverse Laplace transforms. The algorithm was able to predict the temperature distribution of the entire pavement structure by measuring the pavement temperature at a specified time interval and determining the thermal conductivity and diffusivity of the structural layer materials. In the same year, Alawi and Helal [22] proposed

a mathematical model to estimate pavement temperatures using climate data, such as air temperature and solar radiation, to analyze the temperature at different pavement depths. In 2013, Li and Tan [23] established an asphalt pavement analytical model for high temperatures and cold regions, such as seasonally frozen regions, and they analyzed the change law and influencing factors of the asphalt pavement temperature field in typical seasonally frozen regions. In the same year, Wang [24,25] successively proposed approximate classical and infinite-series solutions to analyze the temperature profile of the asphalt pavement layer and compared the analysis results with the measured underground pavement temperature. The results showed that this method could quickly and accurately predict the transient temperature.

In 2014, Chen et al. [26] developed a partial differential equation for one-dimensional heat conduction to simplify the boundary conditions of the pavement temperature field. They derived an explicit expression for the pavement temperature field under specific constraint conditions. Considering the urban heat-island effect [27], an analytical method for predicting the temperature field of asphalt pavements was proposed, and an analytical solution of a multi-layer pavement structure temperature field was derived using the green function method. By using this analytical solution, the temperature field of pavements with different pavement reflectivity's, thermal conductivities, and pavement combinations could be analyzed. In 2016, Qin [28] proposed a theoretical model to predict the pavement surface temperature, which was verified using field data and numerical results from existing studies. It was found that increasing the albedo of the pavement was more effective than increasing its thermal inertia to reduce its surface temperature. In the same year, Dumais and Doré [29] proposed a simplified energy-balance model based on the pavement surface, which was used to calculate the temperature of a high-reflectivity pavement to evaluate its effectiveness. Moreover, a set of design charts based on the radiation index supported by the model was proposed. In 2020, Zhang et al. [30] presented an analytical algorithm for predicting pavement-temperature distributions. The Gauss quadratic formula was applied to the existing concrete pavement system, and an interpolation trigonometric polynomial was used to fit the measured climatic factors in the surface boundary conditions. The temperature solution was verified using measured pavement-temperature data.

Through analysis of the above research results, it is evident that although the theoretical analysis method is complex to solve, it can directly highlight the fundamental principles of the phenomenon being examined and has wide applicability. However, analysis and research on OGFC asphalt pavement temperature fields remain rare. Consequently, a theoretical analysis method was selected in this study to mathematically describe the temperature field of an OGFC asphalt pavement structure based on our knowledge of thermodynamics and mathematical and physical equations and considering the natural environment, characteristics, and properties of an OGFC asphalt pavement structure using mathematical and physical parameters. Based on the separation of variables and the homogenization principle, an analytical equation for the OGFC pavement temperature field was established, and the oil–stone ratio parameter was introduced. The model has three independent variables: the oil–stone ratio, time, and depth.

## 2. Materials and Methods

### 2.1. Materials

OGFC is an open-graded asphalt mixture comprising asphalt and aggregates. Basalt was used as the coarse aggregate, alkaline machine-made sand as the fine aggregate, and ground limestone as the mineral powder. The matrix asphalt was Liaohe 90 # matrix asphalt produced in China, with a PG grade of PG52-28. The technical index of matrix asphalt is illustrated in Table 1. The gradation is illustrated in Figure 1.

**Table 1.** Technical index of matrix asphalt.

| Index | Unit | Test Result |
|---|---|---|
| 25 °C penetration | 0.1 mm | 88.2 |
| Softening point | °C | 49.2 |
| 15 °C ductility | cm | >150 |
| 60 °C viscosity | Pa·S | 228.5 |
| 35 °C viscosity | Pa·S | 0.334 |
| Flash point | °C | 323 |

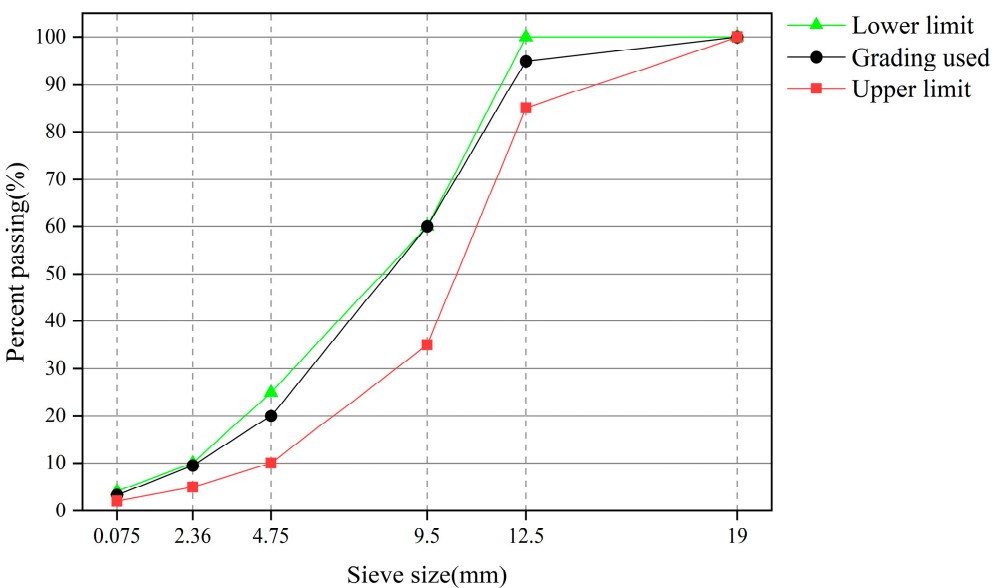

**Figure 1.** Gradation design of open-graded friction course [31].

The oil–stone ratio refers to the percentage of the mass ratio of asphalt to mineral aggregate in asphalt concrete and is one of the indexes of asphalt content. The Marshall test pieces comprised OGFC asphalt mixtures with oil–stone ratios of 5.1%, 5.3%, 5.5%, and 5.7%. Marshall stability tests were conducted to determine whether the mixtures with the above oil–stone ratios met the specification requirements [32]. By measuring the volumes and masses of the Marshall test pieces, the density parameters of the asphalt concrete under these gradations were obtained for subsequent temperature-field calculations.

*2.2. Determination of Thermal Conductivity*

Thermal conductivity can be affected by many factors. Asphalt concrete is a mixture, and its overall thermal conductivity is related not only to the nature of the material itself but also to the proportions of each material component [33]. The thermal conductivity was measured using a plate heat-flow method thermal-conductivity tester.

Test pieces with dimensions of 300 mm × 300 mm × 200 mm (length × width × height) were placed in the thermal-conductivity tester to ensure that the test satisfied the one-dimensional heat-transfer conditions. The thermal conductivities of asphalt concrete at 20, 40, and 60 °C under four different oil–stone ratios were measured; three specimens were set for each oil–stone ratio, and the average values were the thermal conductivity. The results are shown in Figure 2.

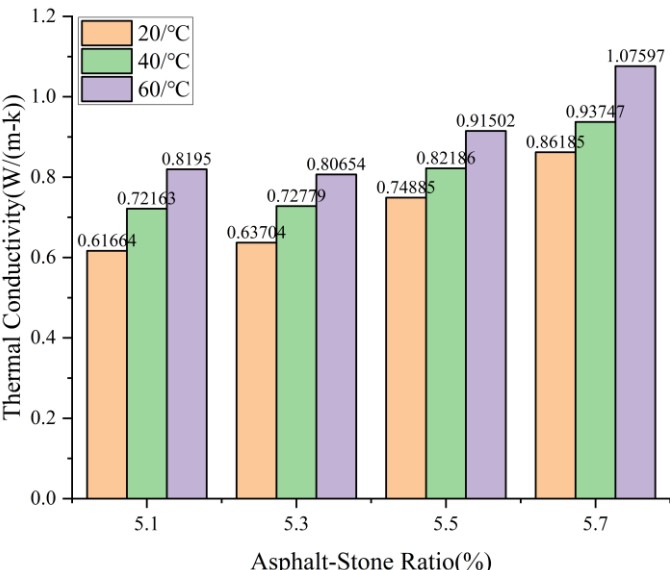

**Figure 2.** Thermal conductivity test results.

It is evident from Figure 2 that there is a relationship between thermal conductivity, oil–stone ratio, and temperature. At the same oil–stone ratio, the thermal conductivity increases with an increase in temperature. At the same temperature, the thermal conductivity increases with an increase in the oil–stone ratio. The change law is close to that of a quadratic polynomial.

Accordingly, the thermal conductivity and oil–stone ratio were mathematically fit, and the test results were formulated. After fitting, the mathematical fitting formula that describes how the thermal conductivity varies with the oil–stone ratio at different temperatures can be obtained, where $\lambda$ is the thermal conductivity (W/(m·k)) and $u$ is the oil–stone ratio (%), with a range of $5.1 \leq u \leq 5.7$. The results are presented in Table 2.

**Table 2.** Curve fit of thermal conductivity to oil–stone ratio.

| Temperature (°C) | Curve Fit of Thermal Conductivity to Oil–Stone Ratio | $R^2$ |
|:---:|:---:|:---:|
| 20 | $\lambda = 15.27579 - 5.82692u + 0.57876u^2$ | 0.96826 |
| 40 | $\lambda = 18.71442 - 7.01762u + 0.68411u^2$ | 0.97849 |
| 60 | $\lambda = 30.17366 - 11.29958u + 1.0869u^2$ | 0.98461 |

## 3. Theory

The temperature field of the pavement structure is its temperature description at any time of day and depth [34]. A temperature-field model of a pavement structure can predict the temperature change and the law of the pavement structure. The following steps were followed to solve the temperature field: the parameters required for the temperature field were determined, the basic temperature-field equation was established, and the conditions to solve the temperature field were determined [21].

### 3.1. Basic Temperature-Field Equation

The temperature-field equation can be used to describe the basic temperature-field laws and is applicable to all heat-conduction situations. Using the differential-element method of thermodynamics, based on the law of conservation of energy and Fourier's law, the general formula for the change in the thermal energy of an object can be derived as follows:

$$\rho c \frac{\partial T}{\partial \tau} = \frac{\partial}{\partial x}\left(\lambda \frac{\partial T}{\partial x}\right) + \frac{\partial}{\partial y}\left(\lambda \frac{\partial T}{\partial y}\right) + \frac{\partial}{\partial z}\left(\lambda \frac{\partial T}{\partial z}\right) + \varphi, \tag{1}$$

where $\rho$ is the density of asphalt concrete (kg/m$^3$); $c$ is the specific heat capacity (J/(kg·°C)); $\tau$ is time (min); $x$, $y$, and $z$ are coordinates of the rectangular coordinate system; $T$ is the temperature difference (°C); $\varphi$ is the calorific value of the internal heat source (kJ).

The internal structure of a traditional road has no heat source or internal heating, and the internal heat source term is zero. Moreover, the establishment of the pavement-structure temperature field must satisfy the following basic assumptions [35]:

1.  The structure of each layer is uniform and homogeneous, showing no obvious difference in either appearance or physical properties at each point.
2.  The cross-sectional temperature at the same depth is the same at different pavement locations, and the heat transfer is only in one dimension, along the longitudinal direction, without considering the transverse distribution of the pavement-structure temperature field and transverse transfer of heat flow.
3.  The materials of each layer of pavement are closely combined, and there is no temperature-field fault; the interlayer temperature and heat flow are continuous; and the heat accumulation phenomenon is not evident and can be ignored.

Based on these basic assumptions, the pavement can be simplified to a one-dimensional longitudinal heat-transfer model [36]. Subsequently, the differential equation for heat conduction in Equation (1) can be simplified as follows:

$$\rho c \frac{\partial T}{\partial \tau} = \frac{\partial}{\partial z}\left(\lambda \frac{\partial T}{\partial z}\right). \tag{2}$$

Substituting the oil–stone ratio equation obtained from Table 1, the oil–stone ratio equation can be expressed by the general expression of a quadratic polynomial:

$$\frac{\partial T}{\partial \tau} = \frac{au^2 + bu + g}{\rho c}\frac{\partial^2 T}{\partial z^2} \tag{3}$$

where $a$, $b$, and $g$ are fitting constants, as follows:

$$\frac{au^2 + bu + g}{\rho c} = d^2. \tag{4}$$

Simplifying Equation (3) to obtain the basic equation of the temperature field yields

$$\frac{\partial T}{\partial \tau} = d^2 \frac{\partial^2 T}{\partial z^2} \tag{5}$$

### 3.2. Boundary Conditions

The aim of this study is to add new boundary conditions and use the separation-variable method and homogenization principle to solve the new analytical solution prediction model of the temperature field. Consequently, we adopted the research results of Yan [37] for the convective heat-transfer boundary conditions, radiation heat-transfer boundary conditions, and temperature change, with the convective heat-transfer boundary conditions expressed as follows:

$$q_1 = (5.7 + 4v)\cdot(T_a - T_r) \tag{6}$$

Here, $T_a = T_1 + T_2[0.96\sin(\omega(\tau - \tau_0)) + 0.14\sin(2\omega(\tau - \tau_0))]$, with $5.7 + 4v$ being a simplified form of the convective heat-transfer coefficient denoted by $B$ (kJ/m$^2$·h·°C); $q_1$ is the convective heat transfer (J); $v$ is the wind speed (m/s); $T_r$ is the air temperature at the road surface (°C); $T_r$ is also the temperature of the road surface (°C) (when $z = 0$), that is, $T(z,\tau)|_{z=0} = T_r$; $T_1$ is the daily average temperature (°C), where $T_1 = (T_{\max} + T_{\min})/2$; $T_2$ is the amplitude of the daily temperature (°C), where $T_2 = (T_{\max} - T_{\min})/2$; $T_{\max}$ is the daily maximum temperature (°C); $T_{\min}$ is the daily minimum temperature (°C); $\tau_0$ is the

initial phase (h), generally taken to be $\tau_0 = 9$; $\omega$ is the angular frequency (rad), $\omega = 2\pi/24$; and $\tau$ is time (h).

We adopted the research results of Huang and Wang [38] for the thermal-radiation boundary conditions, as follows:

$$q_2 = 0.021 Q_d \left(\frac{\bar{t}}{\bar{t}_{\max}}\right)^5 + Q_d \varepsilon^2 \left[0.078\sin\left(\frac{2\pi}{24}(\tau - 6)\right) + 0.034\sin\left(\frac{2\pi}{12}(\tau - 9)\right)\right], \quad (7)$$

where $q_2$ is the radiant heat transfer (J), $Q_d$ is the total daily radiation (kJ/m$^2$), $\varepsilon$ is the radiation-absorption rate of the asphalt pavement or cement pavement (%), $\bar{t}$ is the maximum number of sunshine hours in the month in which the local calculation day occurs (h), $\bar{t}_{\max}$ is the number of maximum sunshine hours in the month with the longest local sunshine time (h), and

*overlinet* and $\bar{t}_{\max}$ are the multi-year averages.

Substituting Equations (6) and (7) into the Fourier law:

$$q = -\lambda \frac{\partial T}{\partial X} \quad (8)$$

we obtain

$$q = q_1 + q_2 = -\lambda \frac{\partial T_r}{\partial X} = B(T_a - T_r) + q_2 \quad (9)$$

Thus,

$$\frac{\partial T_r}{\partial X} - \frac{B}{\lambda} T_r = -\frac{1}{\lambda}(BT_a + q_2). \quad (10)$$

Assuming that $h = \frac{B}{\lambda}$ and $f(\tau) = -\frac{1}{\lambda}(BT_a + q_2)$ and simplifying Equation (10) yields

$$\frac{\partial T_r}{\partial X} - hT_r = f(\tau). \quad (11)$$

When the depth of the pavement structure reaches a certain level, the daily temperature of the pavement structure changes minimally; consequently, the lower boundary condition of the temperature field can be regarded as a constant [39], its temperature is set as a fixed value $e_1$. Using $T(z, \tau)$ to represent the result of the temperature field, the lower boundary condition can be expressed as:

$$T(z, \tau)|_{z=l} = e_1, \quad (12)$$

where $l$ is a certain depth of the pavement at which the daily temperature of the pavement changes slightly (m).

### 3.3. Initial Conditions and Simultaneous Equations

Because the temperature field is affected primarily by the boundary conditions, the impact of the initial conditions on the pavement-structure temperature field is limited to a short period of time when the external temperature starts to change [38], and the daily temperature change of the pavement structure is continuous, rather than starting again every day. Consequently, the initial condition of the temperature field can be considered to be a constant, and $\varphi(z)$ can be expressed in °C. Equations (5), (11), and (12) can be combined to form the following set of equations:

$$\begin{cases} \dfrac{\partial T}{\partial \tau} = d^2 \dfrac{\partial^2 T}{\partial z^2} \\ \dfrac{\partial T_r}{\partial z} - hT_r = f(\tau), \ T(z, \tau)|_{z=l} = e_1. \\ T|_{\tau=0} = \varphi(z) \end{cases} \quad (13)$$

## 4. Determining the Temperature Field

To determine the analytical solution of the temperature field, nonhomogeneous boundary conditions must first be homogenized. It can be assumed that the analytical-solution result consists of two parts, expressed as:

$$T(z, \tau) = V(z, \tau) + B(z, \tau). \tag{14}$$

When $B(z, \tau)$ satisfies

$$B(z, \tau) = \frac{z}{1 + lh}[f(\tau) + he_1] + \frac{1}{1 + lh}[e_1 - lf(\tau)], \tag{15}$$

then $V(z, \tau)$ satisfies the homogenization boundary condition; therefore, it can be assumed that:

$$T(z, \tau) = V(z, \tau) + \frac{z}{1 + lh}[f(\tau) + he_1] + \frac{1}{1 + lh}[e_1 - lf(\tau)]. \tag{16}$$

Substituting Equation (16) into Equation (13), the nonhomogeneous equation group can be resolved into a homogeneous equation group, expressed as:

$$\begin{cases} \frac{\partial V}{\partial \tau} = d^2 \frac{\partial^2 V}{\partial z^2} + \left( \frac{1}{1+lh} - \frac{z}{1+lh} \right) f'(\tau) \\ \left( \frac{\partial V}{\partial z} - hV \right)\Big|_{z=0} = 0, \ V(z, \tau)|_{z=l} = 0 \cdot \\ V|_{\tau=0} = \varphi(z) - B(z, \tau)|_{\tau=0} \end{cases} \tag{17}$$

Based on the physical meaning of the equation, Equation (17) can be decomposed into the temperature field caused by the initial condition, i.e., Equation (18), and that caused by the forced condition, i.e., Equation (19).

The temperature field caused by the initial conditions can be expressed as:

$$\begin{cases} \frac{\partial V_1}{\partial \tau} = d^2 \frac{\partial^2 V_1}{\partial z^2} \\ \left( \frac{\partial V_1}{\partial z} - hV_1 \right)\Big|_{z=0} = 0, \ V_1(z, \tau)|_{z=l} = 0 \cdot \\ V_1|_{\tau=0} = \varphi(z) - B(z, \tau)|_{\tau=0} \end{cases} \tag{18}$$

The temperature field caused by forced conditions can be expressed as:

$$\begin{cases} \frac{\partial V_2}{\partial \tau} = d^2 \frac{\partial^2 V_2}{\partial z^2} + g(z, \tau) \\ \left( \frac{\partial v_2}{\partial z} - hV_2 \right)\Big|_{z=0} = 0, \ V_2(z, \tau)|_{z=l} = 0 \cdot \\ V_2(z, \tau)| = 0 \end{cases} \tag{19}$$

Assuming that

$$g(z, \tau) = \left( \frac{1}{1 + lh} - \frac{z}{1 + lh} \right) f'(\tau), \tag{20}$$

and solving the two parts of the temperature field results, the summation process is $V(z, \tau)$, and the temperature field result $T(z, \tau)$ of the pavement structure can be obtained by substituting $V(z, \tau)$ into Equation (16).

### 4.1. Temperature Field Caused by the Initial Conditions

The result of the temperature field can be determined by the time and depth using the separation-variable method, assuming that

$$V_1(z, \tau) = Z(z)\tau(\tau). \tag{21}$$

Substituting the heat-conduction equation and boundary conditions into Equation (18), we obtain:

$$Z(z)\tau'(\tau) = d^2 Z''(z)\tau(\tau) \tag{22}$$

$$\begin{cases} Z'(0)\tau(\tau) - hZ(0)\tau(\tau) = 0 \\ Z(l)\tau(\tau) = 0 \end{cases}. \tag{23}$$

Introducing variable $-\lambda_n$ to solve Equation (22) yields

$$\frac{\tau'(\tau)}{d^2\tau(\tau)} = \frac{Z''(z)}{Z(z)} = -\lambda_n \tag{24}$$

$$\begin{cases} Z''(z) + \lambda_n Z(z) = 0 \\ \tau'(\tau) + \lambda_n d^2 \tau(\tau) = 0 \end{cases}. \tag{25}$$

The equation set about $Z(z)$ can be established using Equation (23), as follows:

$$\begin{cases} Z''(z) + \lambda_n Z(z) = 0 \\ Z'(0)\tau(\tau) - hZ(0)\tau(\tau) = 0 \end{cases}. \tag{26}$$

The solution can be expressed as:

$$Z_n(z) = B_n\left(\cos\beta_n z + \frac{h}{\beta_n}\sin\beta_n z\right)$$

$$\lambda_n = \beta_n{}^2, \tag{27}$$

where $\beta_n$ is the positive root of $\tan\beta_n l = -\frac{h}{\beta_n}$.

Based on $\tau'(\tau) + \lambda d^2\tau(\tau) = 0$, the equation solution of $\tau(\tau)$ can be obtained:

$$\tau_n(\tau) = A_n e^{-\lambda_n d^2 \tau}. \tag{28}$$

The Fourier series expansion of the equation can be expressed as:

$$V_1(z,\tau) = \sum_{n=1}^{+\infty} C_n e^{-\lambda_n d^2 \tau}\left(\cos\beta_n z + \frac{h}{\beta_n}\sin\beta_n z\right). \tag{29}$$

Substituting the initial conditions based on the generalized Fourier law, we obtain

$$V_1(z,\tau) = \sum_{n=1}^{+\infty} C_n e^{-\lambda_n d^2 \tau}\left(\cos\beta_n z + \frac{h}{\beta_n}\sin\beta_n z\right) = \varphi(z) - B(z,\tau)|_{\tau=0}$$

$$C_n = \frac{\int_0^l [\varphi(z) - B(z,\tau)|_{\tau=0}]\left(\cos\beta_n z + \frac{h}{\beta_n}\sin\beta_n z\right)dz}{\int_0^l [\varphi(z) - B(z,\tau)|_{\tau=0}]^2 dz}. \tag{30}$$

Therefore, the result of the temperature field caused by the initial conditions can be expressed as:

$$V_1(z,\tau) = \sum_{n=1}^{+\infty} C_n e^{-\lambda_n d^2 \tau}\left(\cos\beta_n z + \frac{h}{\beta_n}\sin\beta_n z\right)$$

$$C_n = \frac{\int_0^l [\varphi(z) - B(z,\tau)|_{\tau=0}]\left(\cos\beta_n z + \frac{h}{\beta_n}\sin\beta_n z\right)dz}{\int_0^l [\varphi(z) - B(z,\tau)|_{\tau=0}]^2 dz}. \tag{31}$$

### 4.2. Temperature Field Caused by the Forced Conditions

The temperature field caused by the forced conditions can be determined using the homogenization principle.

It is assumed that

$$V_2(z, \tau) = \int_0^\tau W(z, \tau; m) dm, \tag{32}$$

where $W(z, \tau; m)$ can be solved using Equation (33):

$$\begin{cases} \frac{\partial W}{\partial \tau} = d^2 \frac{\partial^2 W}{\partial z^2} \\ \frac{\partial W}{\partial \tau} - hW \Big|_{z=0} = 0, \ W(z, \tau)|_{z=l} = 0. \\ W(z, \tau)|_{\tau=m} = g(z, m) \end{cases} \tag{33}$$

For the convenience of the solution, the following transformation can be made to Equation (33):

Order: $\tau' = \tau - m$

$$\begin{cases} \frac{\partial W}{\partial \tau'} = d^2 \frac{\partial^2 W}{\partial z^2} \\ \frac{\partial W}{\partial z} - hW \Big|_{Z=0} = 0, \ W(z, \tau)|_{z=l} = 0. \\ W(z, \tau')|_{\tau'=0} = g(z, m) \end{cases} \tag{34}$$

The separation-of-variables method can be used to solve $W(z, \tau; m)$, with the solution process being similar to that used for the initial conditions. By solving $W(z, \tau; m)$, the analytical solution $V_2(z, \tau)$ of the temperature field caused by the forced conditions can be obtained. The final result of $V_2(z\tau)$ can be expressed as:

$$V_2(z, \tau) = \sum_{n=1}^{+\infty} D_n e^{-\lambda_n d^2(\tau - m)} \left( \cos \beta_n z + \frac{h}{\beta_n} \sin \beta_n z \right) dm \tag{35}$$

$$D_n = \frac{\int_0^l g(z, m) \left( \cos \beta_n z + \frac{h}{\beta_n} \sin \beta_n z \right) dz}{\int_0^l \left( \cos \beta_n z + \frac{h}{\beta_n} \sin \beta_n z \right)^2 dz}. \tag{36}$$

The temperature field caused by the initial conditions and that caused by the forced conditions can be summed and sorted, with some parameters being replaced by letters to simplify the written form of the equation. The result can be further resolved by non-integration and substituted into the oil–stone ratio boundary condition. Finally, the analytical solution of the temperature-field equation $T(z, \tau, u)$ can be expressed as:

$$T(z, \tau, u) = \sum_{n=1}^{+\infty} C_n e^{-\lambda_n d^2 \tau} \left( \cos \beta_n z + \frac{h}{\beta_n} \sin \beta_n z \right) + \sum_{n=1}^{+\infty} K e^{-\lambda_n d^2 \tau} \left( \cos \beta_n z + \frac{h}{\beta_n} \sin \beta_n z \right) \{L\} + \frac{z}{1+lh}[f(t) + he_1] + \frac{1}{1+lh}[e_1 - lf(t)] \tag{37}$$

$$\{L\} = M + N + Q \tag{38}$$

$$M = \frac{\frac{1}{\lambda_n d^2} \cdot \frac{\pi}{12} \cdot E \left[ \cos \left( \frac{\pi}{12}(\tau - 9) \right) e^{\lambda_n d^2 \tau} - \cos \left( \frac{\pi}{12}(-9) \right) + \frac{1}{\lambda_n d^2} \cdot \frac{\pi}{12} \sin \left( \frac{\pi}{12}(\tau - 9) \right) e^{\lambda_n d^2 \tau} - \frac{1}{\lambda_n d^2} \cdot \frac{\pi}{12} \sin \left( \frac{\pi}{12}(-9) \right) e^{\lambda_n d^2 \tau} \right]}{1 + \left( \frac{1}{\lambda_n d^2} \right)^2 \cdot \left( \frac{\pi}{12} \right)^2} \tag{39}$$

$$N = \frac{\frac{1}{\lambda_n d^2} \cdot \frac{\pi}{6} \cdot E \left[ \cos \left( \frac{\pi}{6}(\tau - 9) \right) e^{\lambda_n d^2 \tau} - \cos \left( \frac{\pi}{6}(-9) \right) + \frac{1}{\lambda_n d^2} \cdot \frac{\pi}{6} \sin \left( \frac{\pi}{6}(\tau - 9) \right) e^{\lambda_n d^2 \tau} - \frac{1}{\lambda_n d^2} \cdot \frac{\pi}{6} \sin \left( \frac{\pi}{6}(-9) \right) e^{\lambda_n d^2 \tau} \right]}{1 + \left( \frac{1}{\lambda_n d^2} \right)^2 \cdot \left( \frac{\pi}{6} \right)^2} \tag{40}$$

$$Q = \frac{\frac{1}{\lambda_n d^2} \cdot \frac{\pi}{12} \cdot E\left[\cos\left(\frac{\pi}{12}(\tau-6)\right)e^{\lambda_n d^2 \tau} - \cos\left(\frac{\pi}{12}(-6)\right) + \frac{1}{\lambda_n d^2} \cdot \frac{\pi}{12}\sin\left(\frac{\pi}{12}(\tau-6)\right)e^{\lambda_n d^2 \tau} - \frac{1}{\lambda_n d^2} \cdot \frac{\pi}{12}\sin\left(\frac{\pi}{12}(-6)\right)e^{\lambda_n d^2 \tau}\right]}{1 + \left(\frac{1}{\lambda_n d^2}\right)^2 \cdot \left(\frac{\pi}{12}\right)^2} \tag{41}$$

$$C_n = \frac{-\frac{1}{\beta_n}\left(l\sin\beta_n l + \frac{1}{\beta_n}\cos\beta_n l - \frac{1}{\beta_n}\right) + \frac{lh}{\beta_n^2}\left(l\cos\beta_n l - \frac{1}{\beta_n}\sin\beta_n l\right)}{\left(\frac{1}{4\beta_n} - \frac{h^2}{\beta_n^2} \cdot \frac{1}{4\beta_n}\right)\sin 2\beta_n l + \frac{h}{\beta_n^2}\cdot\sin^2\beta_n l + \left(\frac{l}{2} + \frac{h}{\beta_n^2}\cdot\frac{l}{2}\right)} + z\frac{[\varphi(z)-J]\frac{1}{\beta_n}\cdot\sin\beta_n l - [\varphi(z)-J]\frac{h}{\beta_n^2}(\cos\beta_n l - 1)}{\left(\frac{1}{4\beta_n} - \frac{h^2}{\beta_n^2} \cdot \frac{1}{4\beta_n}\right)\sin 2\beta_n l + \frac{h}{\beta_n^2}\cdot\sin^2\beta_n l + \left(\frac{l}{2} + \frac{h}{\beta_n^2}\cdot\frac{l}{2}\right)} \tag{42}$$

$$K = \frac{\left(\frac{1}{lh+1}\right)\frac{1}{\beta_n}\sin\beta_n l - \left(\frac{1}{lh+1}\right)\frac{h}{\beta_n^2}(\cos\beta_n l - 1)}{\left(\frac{1}{4\beta_n} - \frac{h^2}{\beta_n^2} \cdot \frac{1}{4\beta_n}\right)\sin 2\beta_n l + \frac{h}{\beta_n^2}\cdot\sin^2\beta_n l + \left(\frac{l}{2} + \frac{h}{\beta_n^2}\cdot\frac{l}{2}\right)} + \frac{-\left(\frac{1}{lh+1}\right)\frac{1}{\beta_n}\left(l\sin\beta_n l + \frac{1}{\beta_n}\cos\beta_n l - \frac{1}{\beta_n}\right)}{\left(\frac{1}{4\beta_n} - \frac{h^2}{\beta_n^2} \cdot \frac{1}{4\beta_n}\right)\sin 2\beta_n l + \frac{h}{\beta_n^2}\cdot\sin^2\beta_n l + \left(\frac{l}{2} + \frac{h}{\beta_n^2}\cdot\frac{l}{2}\right)} +$$
$$\frac{\left(\frac{1}{lh+1}\right)\frac{h}{\beta_n^2}\left(l\cos\beta_n l - \frac{1}{\beta_n}\sin\beta_n l\right)}{\left(\frac{1}{4\beta_n} - \frac{h^2}{\beta_n^2} \cdot \frac{1}{4\beta_n}\right)\sin 2\beta_n l + \frac{h}{\beta_n^2}\cdot\sin^2\beta_n l + \left(\frac{l}{2} + \frac{h}{\beta_n^2}\cdot\frac{l}{2}\right)} \tag{43}$$

$$I = \frac{f(0) + he_1}{lh + 1} \tag{44}$$

$$J = \frac{e_1 - f(0)}{lh + 1} \tag{45}$$

$$f(0) = E\sin\left(\frac{\pi}{12}(-9)\right) + F\sin\left(\frac{\pi}{6}(-9)\right) + G\sin\left(\frac{\pi}{12}(-6)\right) + H \tag{46}$$

$$E = -\frac{B}{\lambda}\left(\frac{T_{\max} - T_{\min}}{2}\right)\cdot 0.96 \tag{47}$$

$$F = -\frac{B}{\lambda}\left(\frac{T_{\max} - T_{\min}}{2}\right)\cdot 0.14 - \frac{1}{\lambda}\cdot Q_d\cdot\varepsilon^2\cdot 0.034 \tag{48}$$

$$G = -\frac{1}{\lambda}\cdot Q_d\cdot\varepsilon^2\cdot 0.078 \tag{49}$$

$$H = -\frac{B}{\lambda}\left(\frac{T_{\max} - T_{\min}}{2}\right) - \frac{1}{\lambda}\cdot 0.021\cdot\left(\frac{\bar{t}}{\bar{t}_{\max}}\right)^5\cdot Q_d \tag{50}$$

$$\lambda = \beta_n^2. \tag{51}$$

## 5. Verification of the Analytical Solution

To verify the analytical model of the temperature field, a comparison was made using the experimental results. Large (OGFC asphalt mixture) Marshall test pieces were prepared with a diameter and height of 150 mm. Large Marshall specimens were cylindrical; this shape facilitated effective drilling and heat insulation treatment of the specimen. The center of the Marshall specimen is the same distance from the side, which helps to reduce the error caused by side heat transfer. At the same time, the size of large Marshall specimens is larger, which can meet the depth requirement of pavement temperature change research. To make the temperature change obvious, the oil–stone ratio of the OGFC asphalt mixture was 5.7%. According to previous findings, the region where the temperature curve of the pavement structure changes significantly is typically within 0.1 m from the road surface [40]. The thickness of the specimen can be 20–50 mm [41]. In order to ensure that the comparison of data trend changes is more obvious, and the buried depth is within the thickness range of OGFC pavement, holes were drilled into the side of the Marshall test pieces—perpendicular to the longitudinal plane—1, 2, and 3 cm from the upper surface. The holes were positioned

in a straight line, with a hole depth of 75 mm. A temperature sensor was embedded in each hole, after which the holes were sealed with preheated asphalt.

In natural conditions, the plane area of the road surface is quite large, so the heat conduction on the side is nearly negligible. In the traditional study of pavement temperature fields, the transverse heat transfer of pavement is often ignored. Therefore, in order to make the test conditions as close as possible to those of the simulation of the pavement environment and temperature field under natural conditions, it is necessary to insulate the side and bottom of the specimen. The side and bottom surfaces of the Marshall specimens were thermally insulated to approach the temperature field of the pavement structure under natural conditions. After the temperature sensors were embedded in the test pieces, thermal-insulation materials were pasted on the sides and bottom of them to prevent heat loss. The thermal-insulation material comprised three layers of 90-mm thick thermal-insulation cotton with aluminum foil [42]. During the tests, only the upper surfaces of the test pieces could exchange heat with the external environment, similar to the vertical one-dimensional heat transfer of pavement under natural conditions. The heat-insulation process of a test piece is shown in Figure 3.

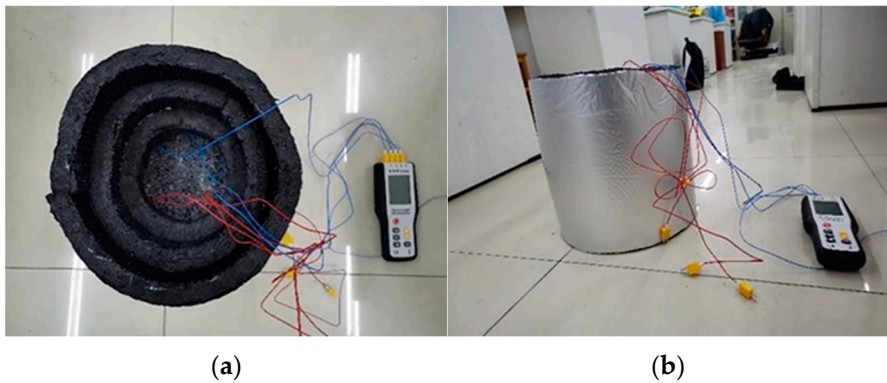

(**a**)　　　　　　　　　　　　　　　　　　　(**b**)

**Figure 3.** Thermal-insulation test piece. (**a**) Top view, and (**b**) side view.

Each fabricated test piece was connected to a temperature sensor in the form of a thermocouple before being placed outdoors under low wind-speed conditions for the test, with the low wind speed simplifying the description of convective heat-transfer conditions [43]. During the test, the radiation received by the test piece was stable, with a low slope, and similar to changes in the primary function. The temperature sensor measured four temperatures simultaneously, that is, the ambient temperature and the temperatures from top to bottom, where the sensors were buried at depths of 1, 2, and 3 cm. The temperature measurements were conducted in real time, with a measurement accuracy within 0.01 °C. The experimental process was as shown in Figure 4.

Specific heat capacity was obtained using the measured data of OGFC in the Shenyang area. When determining the convective heat transfer coefficient, it is first necessary to determine the wind speed of the environment in which the pavement structure is located. For this study, the average daily wind speed level in the Shenyang area was used to calculate the temperature field. The flow heat transfer coefficient of asphalt pavement can be obtained using Equation (6). The density was measured using the volume method (T 0708-2011) in the Standard Test Methods of Bitumen and Bituminous Mixtures for Highway Engineering (JTG E20-2011) [44].

The analytical solution was then substituted using the same specific-parameter values obtained in the temperature-measurement experiment. The data is listed in Table 3.

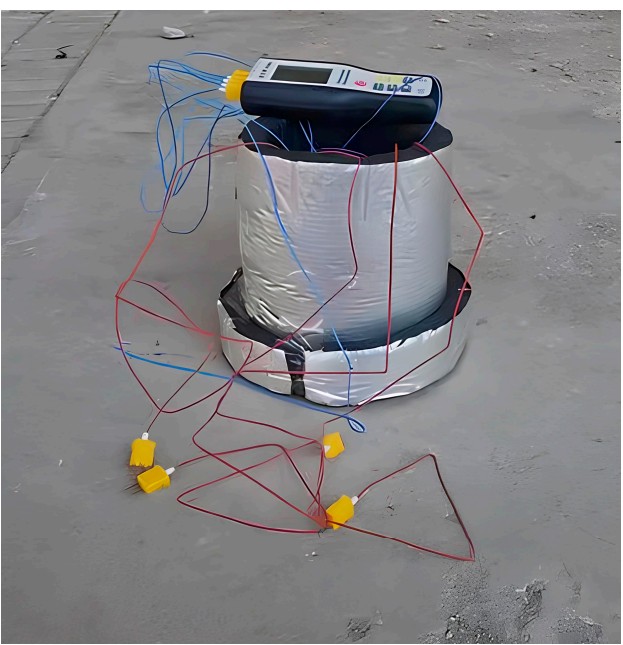

**Figure 4.** Temperature-measurement experiment.

**Table 3.** Selected temperature-field parameters.

| Parameters | Unit | Numerical Value |
|---|---|---|
| Convective heat-transfer coefficient $B$ | kJ/m$^2$·h·°C | 46.024 |
| Density of OGFC asphalt concrete $\rho$ | kg/m$^3$ | 2100 |
| Specific heat capacity of OGFC asphalt concrete $c$ | J/(kg·°C) | 1.0985 |
| Wind speed $v$ | km/h | 1.325 |
| Daily maximum temperature in winter $T_{max}$ | °C | 4 |
| Daily minimum temperature in winter $T_{min}$ | °C | −21 |
| Total daily radiation in winter $Q_d$ | kJ/m$^3$ | 7600 |
| Initial phase $\tau_0$ | h | 9 |
| Maximum sunshine hours in winter months $\bar{t}$ | h | 5.2 |
| Maximum sunshine hours in the longest month $\bar{t}_{max}$ | h | 11.4 |
| Radiation absorptivity $\varepsilon$ | % | 0.87 |
| Underground constant temperature in winter $e_1$ | °C | 5 |
| Initial conditions of temperature field in winter $\varphi(z)$ | - | 1 |

An analytical solution was obtained to predict the results of the model. The test results were then compared with the analytical results, as shown in Figure 5.

It is evident from Figure 5 that the measured data are consistent with the results calculated using the analytical solution, with the same cooling regularity. Through the error bands corresponding to three depths, it can be seen that at 0.01, 0.02, and 0.03 m from the road surface, the maximum differences between the measured data and temperature-field simulation data are 0.5, 0.7, and 0.9 °C, respectively. Consequently, it is evident that the results calculated using the analytical solution are close to those of the actual temperature field.

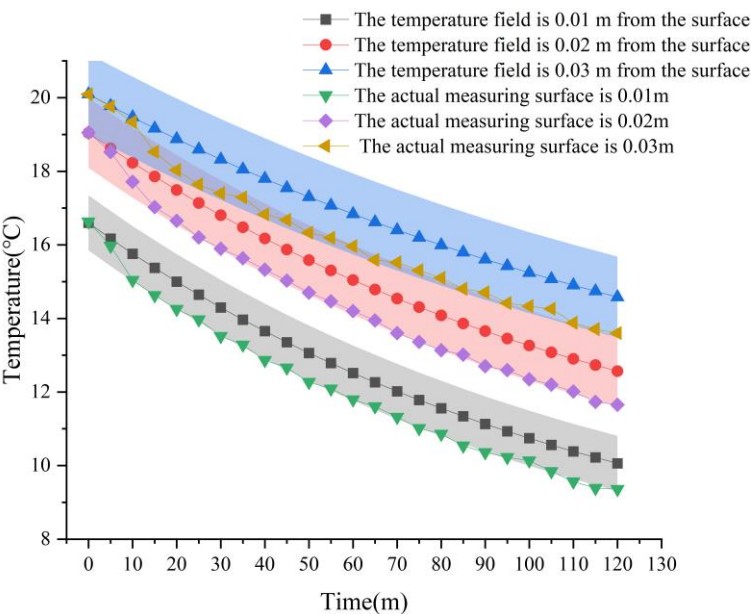

**Figure 5.** Comparison of testing and simulation results.

## 6. Conclusions

Changes in asphalt concrete performance caused by winter icing and summer road softening can increase the risk of traffic accidents. The surface temperature of the road surface must be predicted with high precision so as to optimize traffic control and improve the safety of road driving. In this study, an analytical expression model for the pavement temperature field was established, and the influence of various temperature-field parameters on the pavement temperature change was analyzed through numerical verification and parameter change observation. The main conclusions drawn from this study are as follows:

1.  The relationship between the oil–stone ratio and thermal conductivity of the OGFC asphalt mixture was determined, and a quadratic function was fit to the resulting equation.
2.  Mathematical and physical methods—including the separation of variables and the homogenization principle—were used to solve the temperature field caused by the forced and initial conditions. An analytical solution of the temperature field of the OGFC asphalt pavement structure in the form of a Fourier series was then obtained. The analytical solution of the OGFC asphalt pavement temperature field contained three independent variables: time, depth, and oil–stone ratio.
3.  The analytical solution of the OGFC asphalt pavement temperature field was a composite polynomial comprising exponential and trigonometric functions in the form of a Fourier-series expansion. There were many physical and meteorological parameters in the analytical solution to describe radiation, sunshine, and other changes in the external environment as well as the properties of the pavement structure.
4.  Large Marshall specimens were used as the research objects for the outdoor test. The experimental results were then compared with the analytical-solution prediction model, with the calculated results being the same as those of the actual temperature field.

Potential improvement and further in-depth research are possible on the basis of this study. For example, in order to directly predict the deformation of the road surface, the temperature field and stress field can be coupled to more intuitively examine the impact of the oil–stone ratio on the temperature field and the impact of the temperature field on the stress field. Moreover, in order to investigate the possibility of road surface heat application, it is necessary to combine the research on road surface temperature field and road surface heat application. Further, it is possible to experimentally measure the thermal conductivity of various surface materials, as well as the heat transfer efficiency and heat accumulation

between different layers of asphalt pavement structures and then establish a multi-layer system of pavement temperature field structure.

**Author Contributions:** Conceptualization, B.Y.; Data curation, B.Y.; Formal analysis, B.Y. and Z.Z.; Funding acquisition, L.Q.; Investigation, B.Y. and Z.Z.; Methodology, C.Z.; Project administration, L.Q.; Resources, L.Q. and Z.Z.; Software, Z.Z. and C.Z.; Supervision, L.Q.; Validation, L.Q. and B.Y.; Writing—original draft, L.Q., B.Y. and C.Z.; Writing—review and editing, L.Q. and C.Z. All authors have read and agreed to the published version of the manuscript.

**Funding:** This research was supported by the Scientific Research Development Fund Project of Shenyang Urban Construction University (No. XKJ202309).

**Institutional Review Board Statement:** Not applicable.

**Informed Consent Statement:** Not applicable.

**Data Availability Statement:** The original contributions presented in the study are included in the article; further inquiries can be directed to the corresponding author.

**Acknowledgments:** This study was completed at the School of Civil Engineering in Shenyang Urban Construction University, Transportation Engineering College in Dalian Maritime University, and School of Transportation and Geomatics Engineering in Shenyang Jianzhu University.

**Conflicts of Interest:** The authors declare no conflict of interest.

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
