# Peer review of "Temperature Field Analytical Solution for OGFC Asphalt Pavement Structure"

_coatings, doi:10.3390/coatings13071172_

Round 1

Reviewer 1 Report

1. In figure 2 , how many specimens of each asphalt-stone were prepared? Was there only one or more for each ratio?

Thank you for your careful review and the suggestions. We add the details of the data in Figure 2.

Three specimens were set for each oil–stone ratio, and the average values of were the thermal conductivity.

2. What do you mean by oil-stone ratio? Is it the asphalt volume percent by mass of the aggregates? It needs to be determined early in the text.

Thank you for your careful review and the suggestions. We have improved the introduction of the oil-stone ratio.

The oil–stone ratio refers to the percentage of the mass ratio of asphalt to mineral aggregate in asphalt concrete and is one of the indexes of asphalt content.

3. lines 169-170. It is stated that the conductivity increases with an increase in the oil-stone ratio. This is not he case shown in Figure 2. There is no uniform trend for the variation of the conductivity with the ratio

Thank you for your careful review and the suggestions. After verification, the picture is misplaced and has been corrected.

4. line 372. What are the natural conditions you refer to? Give more details.

Thank you for your careful review and the suggestions. We add the details of natural conditions.

In natural conditions, the plane area of the road surface is quite large, so the heat conduction on the side is nearly negligible. In the traditional study of pavement temperature fields, the transverse heat transfer of pavement is often ignored. Therefore, in order to make the test conditions as close as possible to those of the simulation of the pavement environment and temperature field under natural conditions, it is necessary to insulate the side and bottom of the specimen.

5. line 365. Why did you prepare Marshall specimens with 150mm diameter and height. This needs to be justified.

Thank you for your careful review and the suggestions. We have added the reason for the need for a Marshall specimen with 150mm diameter and heigh in this paper.

Large Marshall specimens were cylindrical; this shape facilitated effective drilling and heat insulation treatment of the specimen. The center of the Marshall specimen is the same distance from the side, which helps to reduce the error caused by side heat transfer. At the same time, the size of large Marshall specimen is larger, which can meet the depth requirement of pavement temperature change research.

6. line 390. The sensors were buried at depths of 1, 2 and 3 cm. Why did you consider these depths? What is the thickness of the OFGC layer on a constructed pavement?.

Thank you for your careful review and the suggestions. We insert the relevant literature in the text to explain the thickness of the OFGC layer on the built road surface and the reason for considering the buried depth of the sensor.

According to previous findings, the region where the temperature curve of the pavement structure changes significantly is typically within 0.1 m from the road surface [39]. The thickness of the specimen can be 20–50 mm [40]. In order to ensure that the comparison of data trend changes is more obvious, and the buried depth is within the thickness range of OGFC pavement,

[39] GANCHANG W U. The analytic theory of the temperature fields of bituminous pavement over semi-rigid roadbase[J]. Applied Mathematics and Mechanics, 1997,18: 181-190.

[40] WANG Y, LENG Z, WANG G. Structural contribution of open-graded friction course mixes in mechanistic–empirical pavement design[J]. International Journal of Pavement Engineering, 2014,15: 731-741.

7. Table 2. Please explain how you came up with these values. Were these values measured or assumed from relevant studies or based on bibliography?.

Thank you for your careful review and the suggestions. We explain the source of the relevant data in Table 2 in the text.

8. More explanation is needed since you recorder the temperature but other parameters are needed, such as convective heat-transfer coefficient, specific heat capacity etc.

Thank you for your careful review and the suggestions. We added the explanation of convective heat-transfer coefficient, specific heat capacity in the text as following:

Specific heat capacity was obtained using the measured data of OGFC in the Shenyang area. When determining the convective heat transfer coefficient, it is first necessary to determine the wind speed of the environment in which the pavement structure is located. For this study, the average daily wind speed level in Shenyang area was used to calculate the temperature field. The flow heat transfer coefficient of asphalt pavement can be obtained using Equation (6).

Here, , with  being a simplified form of the convective heat-transfer coefficient denoted by B (kJ/m2·h·â„ƒ)

9. How did you calculate the density of the OFC presented in table 2?

Thank you for your careful review and the suggestions. We add the correlation of OGFC density calculation in this paper.

The density was measured using the volume method (T 0708-2011) in the Standard Test Methods of Bitumen and Bituminous Mixtures for Highway Engineering (JTG E20-2011) [43].

[43] Standard Test Methods of Bitumen and Bituminous Mixtures for Highway Engineering, China communications press, 2011.

10. the shortcoming of the proposed method should be presented and discussed.

Thank you for your careful review and the suggestions. We have added the proposal and discussion of the shortcomings of our method in the conclusion.

Failure to consider the heat transfer efficiency and heat accumulation between different layers of asphalt pavement structures, as well as the differences in properties between different materials. Failure to establish a multi-layer pavement temperature field structure.

11. Also, the aspects for further investigation should be identified.

Thank you for your careful review and the suggestions. We increase the prospect of this article in the conclusion.

Potential improvement and further in-depth research are possible on the basis of this study. For example, in order to directly predict the deformation of the road surface, the temperature field and stress field can be coupled to more intuitively examine the impact of the oil–stone ratio on the temperature field and the impact of the temperature field on the stress field. Moreover, in order to investigate the possibility of road surface heat application, it is necessary to combine the research on road surface temperature field and road surface heat application. Further, it is possible to experimentally measure the thermal conductivity of various surface materials, as well as the heat transfer efficiency and heat accumulation between different layers of asphalt pavement structures and then establish a multi-layer system of pavement temperature field structure.

12. There should be a discussion point on the importance of predicting the temperature with great accuracy of the pavement surface layer.

Thank you for your careful review and the suggestions. We add a discussion point about high-precision prediction of pavement surface temperature.

Changes in asphalt concrete performance caused by winter icing and summer road softening can increase the risk of traffic accidents. The surface temperature of the road surface must be predicted with high precision so as to optimize traffic control and improve the safety of road driving.

13. Authors need to show how the results of their research can be utilised in pavements.

Thank you for your careful review and the suggestions. We add in the conclusion how this study is applied to practice.

In this study, an analytical expression model for the pavement temperature field was established, and the influence of various temperature-field parameters on the pavement temperature change was analyzed through numerical verification and parameter change observation.

Reviewer 2 Report

The reviewer thanks for the authors efforts. Several very important comments should be addressed:

1. Please revise the title. It includes five nouns in a row which grammatically is incorrect.

Thank you for your careful review and the suggestions. We modified the title of the paper.

Temperature Field Analytical Solution for OGFC Asphalt Pavement Structure

2. Line 35-37. Please refer also to the drawbacks such as increased ageing susceptibility.

Thank you for your careful review and the suggestions. We add a reference to the aging sensitivity of OGFC.

In addition, sunlight, air, and water can easily enter the OGFC interior, accelerating the asphalt aging process and causing durability problems [6].

[6]   TIAN X, HUANG X, YU B. Experimental Study on the Durability of Open Graded Friction Course Mixtures, 2011.

3. Section 2.1. What type of asphalt binder was used? Please provide info and classification of it i.e. based on PG.

Thank you for your careful review and the suggestions. We have increased the relevant indicators of asphalt

The matrix asphalt was Liaohe 90 # matrix asphalt produced in China, with a PG grade of PG52-28. The echnical index of matrix asphalt is illustrated in Table 1.

 Table 1. Technical index of matrix asphalt

Index

Unit

Test result

25 °C penetration

0.1 mm

88.2

Softening point

°C

49.2

15 °C ductility

cm

>150

60 °C viscosity

Pa·S

228.5

35 °C viscosity

Pa·S

0.334

Flash point

°C

323

4. Section 2.2. belongs to the results.

Thank you for your careful review and the suggestions. The determination of thermal conductivity is to lay the foundation for solving the analytical solution of the temperature field later. The heat conduction test and data fitting need to be placed before the analytical solution of the temperature field.

5. Line 207. For similar simplified modelling approaches you can consult and refer to https://doi.org/10.1016/j.conbuildmat.2020.120592.

Thank you for your careful review and the suggestions.

We introduce the literature in the text to increase the persuasiveness of the article.

[35] MÜHLICH U, PIPINTAKOS G, TSAKALIDIS C. Mechanism based diffusion-reaction modelling for predicting the influence of SARA composition and ageing stage on spurt completion time and diffusivity in bitumen[J]. Construction and Building Materials, 2020: 120592.

6. Line 410-41. Please critically evaluate the obtained results and the convergence of modelling and experimental data, especially since other approaches offer better accuracy.

Thank you for your careful review and the suggestions. We have modified Figure 5 as a strip error plot to improve the accuracy of the paper.

Figure 5. Comparison of test with the simulation results.

Round 2

Reviewer 1 Report

none

please review all article for English editing